# Tropical land use alters functional diversity of soil food webs and leads to monopolization of the detrital energy channel

Zheng Zhou[1]*, Valentyna Krashevska[1], Rahayu Widyastuti[2], Stefan Scheu[1,3], Anton Potapov[1,4]

[1]JF Blumenbach Institute of Zoology and Anthropology, University of Göttingen, Göttingen, Germany; [2]Department of Soil Sciences and Land Resources, Institut Pertanian Bogor, Bogor, Indonesia; [3]Centre of Biodiversity and Sustainable Land Use, Göttingen, Germany; [4]Russian Academy of Sciences, A.N. Severtsov Institute of Ecology and Evolution, Moscow, Russian Federation

*For correspondence:
zzhou@gwdg.de

Competing interest: The authors declare that no competing interests exist.

**Abstract** Agricultural expansion is among the main threats to biodiversity and functions of tropical ecosystems. It has been shown that conversion of rainforest into plantations erodes biodiversity, but further consequences for food-web structure and energetics of belowground communities remains little explored. We used a unique combination of stable isotope analysis and food-web energetics to analyze in a comprehensive way consequences of the conversion of rainforest into oil palm and rubber plantations on the structure of and channeling of energy through soil animal food webs in Sumatra, Indonesia. Across the animal groups studied, most of the taxa had lower litter-calibrated $\Delta^{13}C$ values in plantations than in rainforests, suggesting that they switched to freshly-fixed plant carbon ('fast' energy channeling) in plantations from the detrital C pathway ('slow' energy channeling) in rainforests. These shifts led to changes in isotopic divergence, dispersion, evenness, and uniqueness. However, earthworms as major detritivores stayed unchanged in their trophic niche and monopolized the detrital pathway in plantations, resulting in similar energetic metrics across land-use systems. Functional diversity metrics of soil food webs were associated with reduced amount of litter, tree density, and species richness in plantations, providing guidelines on how to improve the complexity of the structure of and channeling of energy through soil food webs. Our results highlight the strong restructuring of soil food webs with the conversion of rainforest into plantations threatening soil functioning and ecosystem stability in the long term.

## Editor's evaluation

Zhou et al., provide a robust study on isotopic and metabolic changes of a soil community across a gradient of different land-use types in Sumatra, Indonesia. By mixing community-based analyses of stable isotopes and size-based metabolic measures, they are able to elucidate, for the first time, important links among plants and the soil food web in tropical ecosystems. This study is of importance to tropical biologists, ecosystem ecologists and biodiversity conservationists aiming to understand the impacts of humans on tropical forests.

## Introduction

Worldwide, land use changes the structure of ecological communities and is associated with losses in multiple ecosystem functions, which is at the core of sustainable development goals (*Bommarco et al., 2013*; *Matson et al., 1997*; *Newbold et al., 2015*). Many tropical ecosystems are affected by land-use changes, losing their biodiversity and multifunctionality (*Barnes et al., 2014*; *Laurance, 2007*). It is projected that tropical ecosystems will face even greater pressures due to land-use change in the future (*Dobrovolski et al., 2011*). Decreases in biodiversity and changes in trophic interactions in animal communities (*Newbold et al., 2015*; *Tsiafouli et al., 2015*; *Wilkinson et al., 2021*) are associated with changes in nutrient dynamics and energy fluxes (*de Vries et al., 2012*; *McGrath et al., 2001*; *Potapov et al., 2020*), which ultimately influence ecosystem functioning and stability (*Rooney et al., 2006*; *Rooney and McCann, 2012*). However, interrelationships between the loss of diversity and changes in energy pathways in food webs are poorly studied and this applies in particular to tropical ecosystems.

Soils harbor a large portion of terrestrial biodiversity (*Guerra et al., 2021*), are intimately linked with aboveground biodiversity (*Bardgett and Putten, 2014*; *Hooper et al., 2000*; *Yang et al., 2018*), and deliver vital ecosystem services (*Bardgett and Wardle, 2010*; *de Vries et al., 2013*). Energetically, 80–90% of the carbon fixed by plants in terrestrial ecosystems enters the belowground system (*Gessner et al., 2010*) and is processed in soil food webs by microorganisms and invertebrate decomposers, the latter then become prey for predators (*Bardgett and Wardle, 2010*; *Schmitz and Leroux, 2020*). Shifts in resource use in the decomposer system results in asymmetries in energy fluxes through soil food-web channels, which modulate the resistance and resilience of terrestrial ecosystems to perturbations (*de Vries et al., 2006*; *de Vries et al., 2012*; *Rooney et al., 2006*; *Rooney and McCann, 2012*). Studies in temperate regions showed that more intensive land use reduces the diversity of soil organisms (*Tsiafouli et al., 2015*) and shifts soil food webs toward the 'fast' bacterial energy channel at the expense of the 'slow' fungal energy channel (*de Vries et al., 2006*), potentially undermining food-web stability. However, knowledge on how the rapid land-use change in tropical regions, such as the conversion of rainforest into plantations, affects soil food-web structure and energy channeling is scarce (*Clough et al., 2016*; *Dobrovolski et al., 2011*).

The present study took place in Jambi province, Sumatra, Indonesia, which is a global hotspot of biodiversity (*Koh and Ghazoul, 2010*; *Miettinen et al., 2011*), where over last 25–35 years rainforests and agroforests have been largely replaced by intensively managed plantations, mostly oil palm and rubber (*Clough et al., 2016*; *Margono et al., 2012*). Results of previous studies showed that land-use change in this region is associated with changes in soil chemistry (*Ballauff et al., 2021*), shifts in microbial and plant communities (*Krashevska et al., 2015*; *Rembold et al., 2017a*; *Schulz et al., 2019*), and reduced multitrophic biodiversity and functionality of soil animal communities (*Barnes et al., 2014*; *Krause et al., 2021*; *Potapov et al., 2019a*; *Krashevska et al., 2019*). These changes are expected to affect trophic niches of soil animals and alter both structure and energetics of soil food webs. In certain soil invertebrate groups (e.g., centipedes, springtails, and mites), conversion of rainforest into plantation systems has been associated with trophic shifts toward the plant energy channel in plantations (*Klarner et al., 2017*; *Krause et al., 2021*; *Susanti et al., 2021*), whereas the bacterial channel was reduced, as suggested by fatty acid analysis (*Susanti et al., 2019*). A more complete assessment of soil food webs showed that plantations are energetically dominated by large decomposers (i.e., earthworms), but have largely reduced energy fluxes to predators (*Barnes et al., 2014*; *Potapov et al., 2019a*), however, these studies ignored potential shifts in the trophic niches of individual soil taxa with land-use change.

Progress in understanding soil food-web responses to environmental changes is hampered by the chronic lack of empirical data for complex soil food webs (*Brose and Scheu, 2014*). Insights in their structure became possible with introduction of stable isotope, molecular, and biochemical methods, which showed inaccuracies in the traditional reconstructions (*Bradford, 2016*; *Brose and Scheu, 2014*; *Geisen et al., 2019*). Stable isotope analysis is now widely used as a first-line explorative tool in trophic ecology (*Peterson and Fry, 1987*; *Parnell et al., 2010*), allowing for in situ assessment of soil food-web structure (*Potapov et al., 2019c*). The method is especially promising to provide insight into the structure of soil food webs in the tropics, where the biology of species is poorly known. The $^{13}C/^{12}C$ and $^{15}N/^{14}N$ ratios in consumers depend on their food and can be used to explore the trophic niches of animal species and communities (*Post, 2002a*; *Pollierer et al., 2009*; *Potapov*

*et al., 2019c*). The $^{15}N/^{14}N$ isotope ratio is used to indicate the trophic position of species since it is enriched by about 3–4‰ per trophic level (*Post, 2002a*; *Pollierer et al., 2009*; *Potapov et al., 2019c*); $^{13}C$ typically is little affected by trophic transfer and thus reflects basal food resources of the trophic chain (*Peterson and Fry, 1987*; *Potapov et al., 2019c*). In soil communities, animals with high $^{13}C$ concentration are considered to use 'older' carbon that has higher $^{13}C$ values due to decomposition processes and preferential incorporation of labile plant compounds by microbes (*Pollierer et al., 2009*; *Potapov et al., 2019c*), and those with lower $^{13}C$ concentration are considered to feed on freshly fixed plant material (*Fujii et al., 2021*; *Potapov et al., 2019c*).

To assess food-web structure using stable isotope analysis, *Layman et al., 2012*; *Layman et al., 2007*, suggested a number of 'isotopic metrics', which have been widely used in aquatic ecology. These metrics consider all species as having the same importance for food-web structure, which has a binary perspective (i.e., presence/absence of species), ignoring potential asymmetries in the magnitude of trophic interactions. However, in biological communities often only few species dominate, forming the energetic core of the food web, therefore, a non-binary perspective is important. Recently, *Cucherousset and Villéger, 2015*, joined Layman's metrics and the functional diversity framework (*Petchey and Gaston, 2006*; *Villéger et al., 2008*; *Mouillot et al., 2013*) to calculate functional diversity indices for food webs, accounting for the dominance of species. These indices include isotopic 'richness' which represents the volume of the trophic niche across all species, isotopic 'evenness' which represents the regularity of the distribution of species' trophic niches, isotopic 'divergence' which reflects the dominance of species with the most extreme trophic niches, and isotopic 'dispersion' which reflects the balance of the species distribution in the trophic space (*Cucherousset and Villéger, 2015*). These isotopic indices represent basic components of the functional diversity of food webs. Nevertheless, to our knowledge they have never been used to analyze soil food-web characteristics, either temperate or tropical, except for one case study on oribatid mites (*Krause et al., 2021*). Moreover, abundance and biomass each are biased in reflecting the functional role of consumers covering wide body size ranges. While abundance is biased toward the importance of small organisms, biomass is biased toward that of large ones. Considering these limitations, energetic demands of consumers (i.e., metabolic rates) may be used as less biased metric (*Brown et al., 2004*). In recent years, the energy flux approach was successfully used to represent functional changes in food webs, and therefore to link multitrophic biodiversity to ecosystem functioning (*Barnes et al., 2018*; *Barnes et al., 2014*; *Jochum et al., 2021*). To the best of our knowledge, however, the energy flux approach has never been used in conjunction with stable isotope analysis.

Here, for the first time we use stable isotope analysis to comprehensively investigate changes in tropical soil food webs associated with changes in land use. We apply a functional diversity framework to stable isotope data to assess which structural dimensions of soil food webs vary most across rainforests, agroforests, and intensively managed plantations of oil palm and rubber in Jambi province, Sumatra, Indonesia (*Clough et al., 2016*; *Drescher et al., 2016*). Using data on 23 high-rank taxonomic groups (orders, families), we focus on two perspectives of the functional diversity of soil food webs: a 'community perspective' in which we treat all groups as being equally important and an 'energetic perspective' in which we weight groups according to their shares in community metabolism. For both perspectives, we tested the following hypotheses: (1) shifts in trophic niches are uniform across all studied animal groups through land-use changes, with animals in plantations being less enriched in $^{13}C$ than in rainforest due to stronger plant and weaker detrital energy channel; (2) functional diversity of soil food webs declines with land-use intensity in plantation systems reflected by reduced isotopic richness, redundancy, evenness, and divergence; (3) from an 'energetic perspective' soil food webs are less affected by changes in land use than from a 'community perspective' as total energy flux changes little with conversion of rainforest into plantations, whereas biodiversity declines strongly. Lastly, we aim at identifying the environmental factors driving changes in functional diversity of soil food webs under land-use change, both from a community and energetic perspective.

## Results

### Isotopic shifts in individual animal taxa

If averaged across rainforest sites, the mean $\Delta^{13}C$ values of taxonomic groups covered a range of 3.0‰, from 3.4‰ (Coleoptera and Oribatida) to 6.4‰ (Orthoptera and Pauropoda). The respective

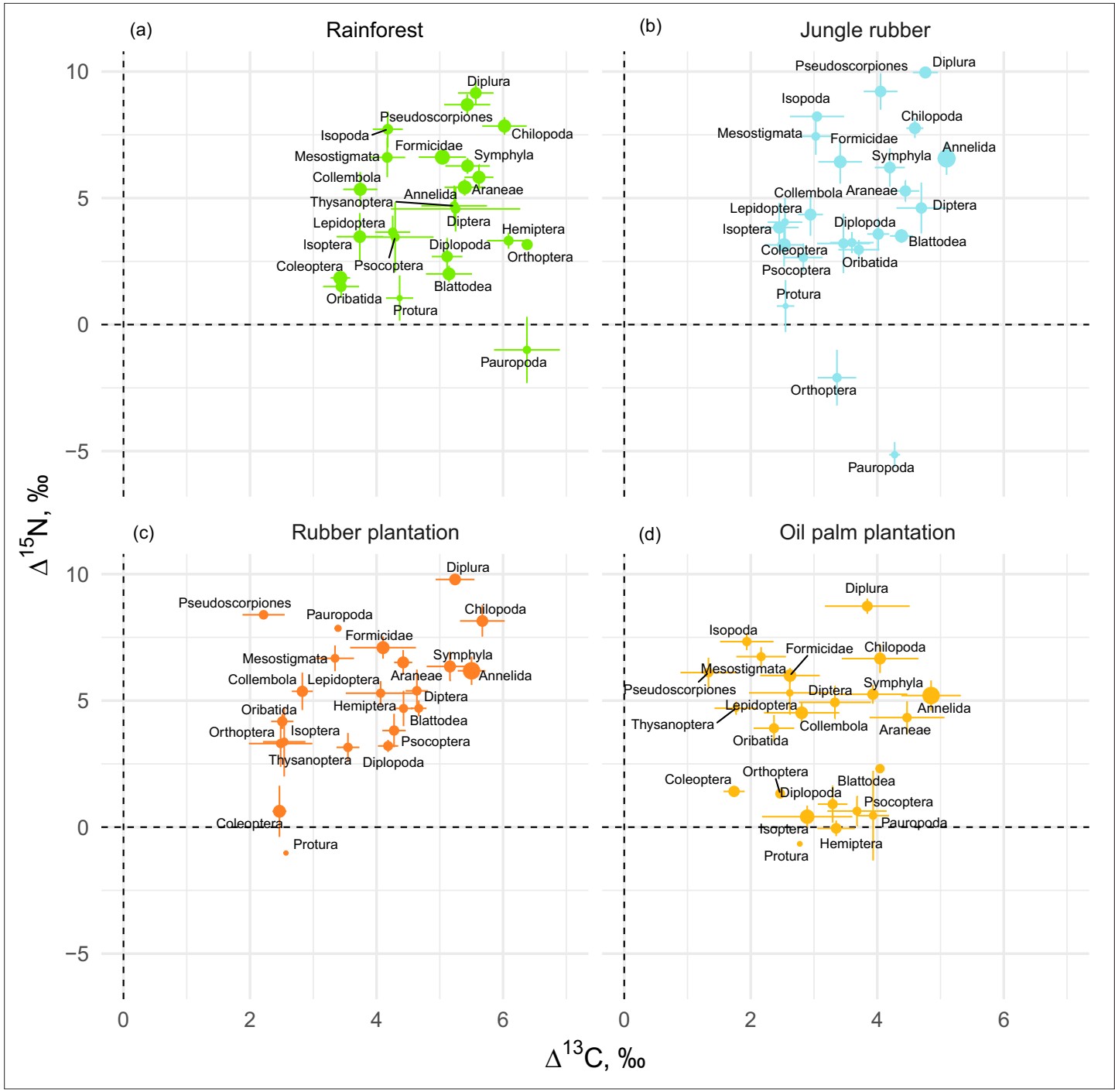

**Figure 1.** Mean litter-calibrated $\Delta^{13}C$ and $\Delta^{15}N$ values of soil animal taxa in rainforest (**a**), jungle rubber (**b**), rubber, (**c**) and oil palm plantations (**d**). Error bars represent standard errors across sampling plots (n = 1–8 per land-use system). Size of the points is scaled to the total share of the taxonomic group in the community metabolism in the corresponding land-use system (metabolism was $\log_{10}$-transformed to show trends in rarer groups).

The online version of this article includes the following source data and figure supplement(s) for figure 1:

**Source data 1.** Metabolism data of each group in each plot.

**Source data 2.** Stable isotope data of groups in each plot.

**Figure supplement 1.** Metabolism proportion of animal groups in different land-use systems.

$\Delta^{15}$N values covered a range of 14.2‰, from –5‰ (Pauropoda) to 9.2‰ (Diplura; *Figure 1a*). The $\Delta^{13}$C values of all groups varied across the four land-use systems, but Chilopoda, Diplura, and Annelida were typically most enriched in $^{13}$C and Coleoptera were consistently among the most depleted in $^{13}$C among all groups. Diplura, Pseudoscorpiones, Chilopoda, and Isopoda had the highest $\Delta^{15}$N values among all groups across the four land-use systems. The group with lowest $\Delta^{15}$N values was Pauropoda in rainforest and jungle rubber, and Protura in rubber and oil palm plantations. Overall, micropredators, that is, Diplura and Pseudoscorpiones, had 2–3‰ higher $\Delta^{15}$N values than macropredators, that is, Chilopoda, Formicidae, and Araneae (*Figure 1*). The share of Annelida (earthworms) in community metabolism was 15.4% in rainforest, but represented more than 75% in jungle rubber, rubber, and oil palm plantations (*Figure 1—figure supplement 1*).

The $\Delta^{13}$C values were significantly higher in rainforest than in the other land-use systems in Coleoptera, Diplopoda, Hemiptera, Orthoptera, Pauropoda, Protura, Pseudoscorpiones, and Thysanoptera (*Figure 2*, *Figure 2—figure supplement 1*). Chilopoda, Diplura, Formicidae, Isopoda, Mesostigmata, and Symphyla were significantly more enriched in $^{13}$C in rainforest than in oil palm, but not significantly different from those in jungle rubber and rubber plantations. In general, most groups in rainforest were higher in $\Delta^{13}$C by 1–3‰ than in the other land-use systems, but this shift was only significant for two out of six macrodecomposer groups. Annelida, which accounted for much of the community metabolism in each of the land-use systems, had similar $\Delta^{13}$C values across land-use systems.

The $\Delta^{15}$N values were by 1.5–2.5‰ lower in rainforest than in the other land-use systems in Oribatida and Blattodea (except for similar $\Delta^{15}$N values in oil palm and rainforest for Blattodea). By contrast, $\Delta^{15}$N values of Hemiptera, Orthoptera, Isoptera, and Pseudoscorpiones were lower in oil palm than in rainforest, whereas in jungle rubber this was only true for Orthoptera (*Figure 2*, *Figure 2—figure supplement 2*).

## One-dimensional isotopic metrics

One-dimensional isotopic metrics described the overall range and average $\Delta^{13}$C and $\Delta^{15}$N values of each community. The maxima of $\Delta^{13}$C values were by 1–2‰ higher in forest than in jungle rubber and oil palm plantations, but minima and the overall range did not differ significantly (*Figure 3*). The unweighted average $\Delta^{13}$C values of communities were by 1–2‰ higher in rainforest than in the other land-use systems and were also higher in rubber than in oil palm plantations. However, the energetic average positions did not differ significantly due to similar $\Delta^{13}$C values of Annelida (dominant invertebrate group) across land-use systems (*Figure 2e*, *Figure 1—figure supplement 1*).

Extreme values of $\Delta^{15}$N were most pronounced in jungle rubber, both maximum and minimum, resulting in the largest range in $\Delta^{15}$N values (16.5‰) among the four land-use systems. In the other land-use systems, maxima, minima, and ranges of $\Delta^{15}$N values were similar. The unweighted average $\Delta^{15}$N values of communities were lowest in oil palm, being significantly lower than in rubber (*Figure 3i*). By contrast, the energetic average $\Delta^{15}$N values did not differ significantly.

## Multidimensional isotopic metrics

Among the unweighted metrics (i.e., community metrics), isotopic dispersion was significantly higher in oil palm than in each of the other land-use systems; isotopic divergence and uniqueness were significantly higher in oil palm than in jungle rubber; isotopic evenness was significantly lower in jungle rubber than in each of the other land-use systems; only isotopic richness showed no significant differences between land-use systems, but in trend the two monoculture systems had lower values than in rainforest and jungle rubber. For detailed information on the plot-level metrics values, see Appendix (*Figure 4—figure supplements 1 and 2*).

By contrast, the weighted multidimensional metrics (i.e., energetic metrics) did not differ among land-use systems for isotopic dispersion, isotopic evenness, isotopic richness, and isotopic uniqueness (*Figure 4*). Only isotopic divergence was significantly lower in rainforest than in the other land-use systems, showing an opposite trend to isotopic dispersion. For detailed information on plot level metrics values see Appendix (*Figure 4—figure supplements 3–6*).

## Environmental effects on functional diversity of soil food webs

As indicated by multivariate analysis, community metrics differed strongly between the four land-use systems (anosim R = 0.404, p < 0.001), whereas differences for the energetic metrics were less

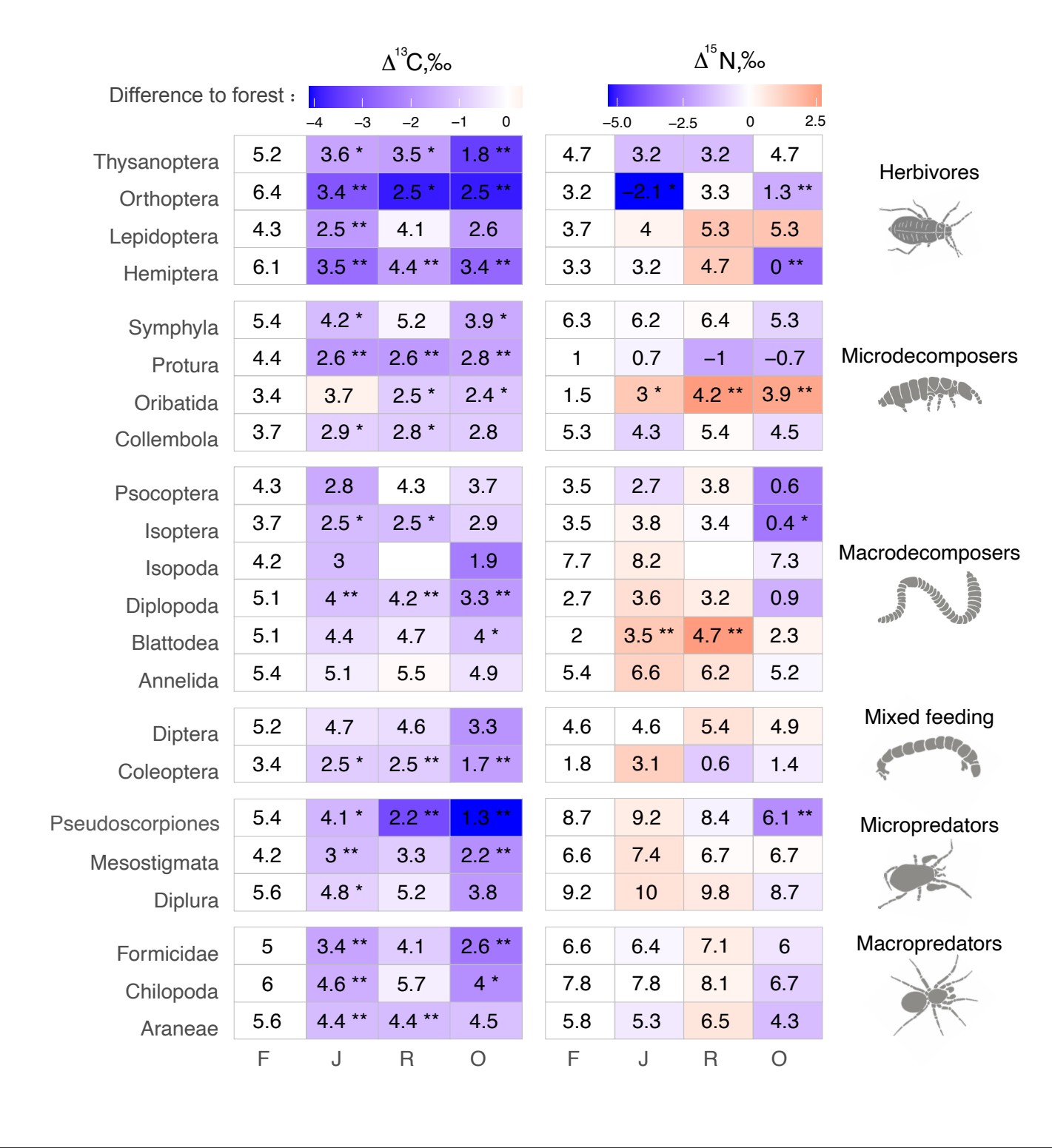

**Figure 2.** Average Δ¹³C and Δ¹⁵N values of taxonomic groups in rainforest (**F**), jungle rubber (**J**), rubber, (**R**) and oil palm plantations (**O**). Numbers show means, asterisks indicate significant differences between the mean value in the corresponding land-use system and in rainforest (Student's t-test *p < 0.05, **p < 0.01). Color represents the direction (red – increase, blue – decrease) and magnitude (darker color indicate stronger change) of the difference between rainforest and other land-use systems.

The online version of this article includes the following source data and figure supplement(s) for figure 2:

*Figure 2 continued on next page*

*Figure 2 continued*

**Source data 1.** Metabolism data of each group in each plot.

**Figure supplement 1.** Δ13C of each taxa in in rainforest (F, green), jungle rubber (J, blue), rubber (R, red), and oil palm plantations (O, yellow).

**Figure supplement 2.** Δ15N of each taxa in in rainforest (F, green), jungle rubber (J, blue), rubber (R, red), and oil palm plantations (O, yellow).

pronounced (anosim R = 0.138, p = 0.014). Among all tested environmental factors, soil pH, tree species richness, litter amount, and understory density had the strongest correlations with both community metrics (p = 0.003, p = 0.037, p < 0.001, p < 0.001, respectively) and energetic metrics (p = 0.004, p = 0.002, p = 0.003, p = 0.002, respectively). These variables were subsequently selected for the structural equation model (SEM) analysis (see Materials and methods). SEM indicated that the changes in the community metrics (PC1$_{unweighted}$) were induced directly by tree properties and litter amount (tree density: p < 0.05, effect size = 0.72; tree species richness: p < 0.001, effect size = –0.83; litter amount: p < 0.001, effect size = 0.71), while changes in the energetic metrics (PC1$_{weighted}$) were indirectly driven by soil pH via increased metabolism of earthworms (p < 0.05, effect size = –0.36; *Figure 5*; *Figure 6*).

## Discussion

We used stable isotope data of 23 high-rank animal taxa to comprehensively assess changes in functional diversity of soil food webs under tropical land-use change. We found shifts in basal resource use for most taxonomic groups in plantations compared to rainforest, and responses of food-web

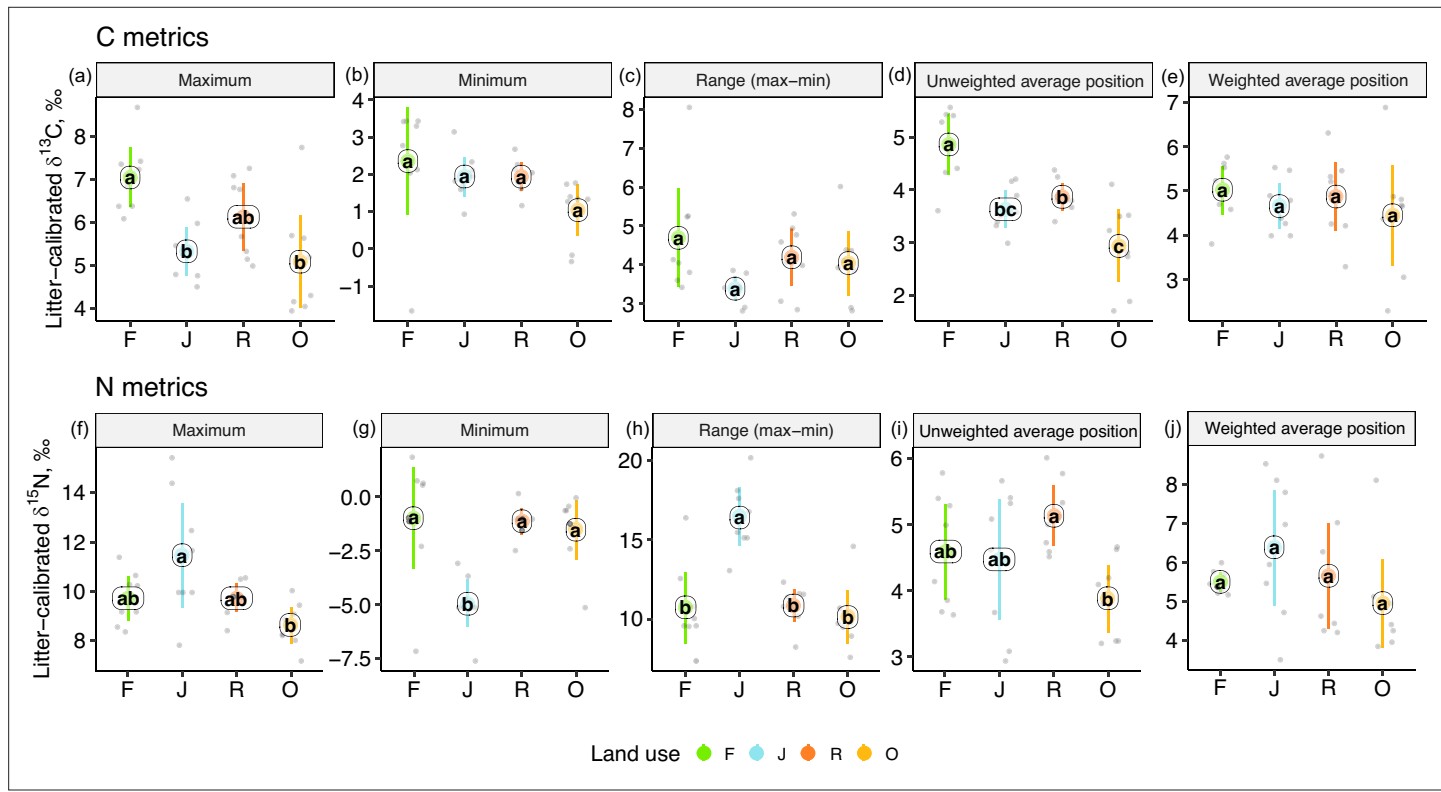

**Figure 3.** One-dimensional metrics for Δ13C (upper panel) and Δ15N values (lower panel) of communities in rainforest (F, green), jungle rubber (J, blue), rubber (R, red), and oil palm plantations (O, yellow). Each point represents one community (n = 8 per land-use system). For the calculation of the weighted average values, species were weighted according to their contribution to the total community metabolism per plot. Means sharing the same letter within each pane are not significantly different (Tukey's HSD test following ANOVA, p > 0.05).

The online version of this article includes the following source data for figure 3:

**Source data 1.** Community metrics of soil food webs in each plot.

**Source data 2.** Energetic metrics of soil food webs in each plot.

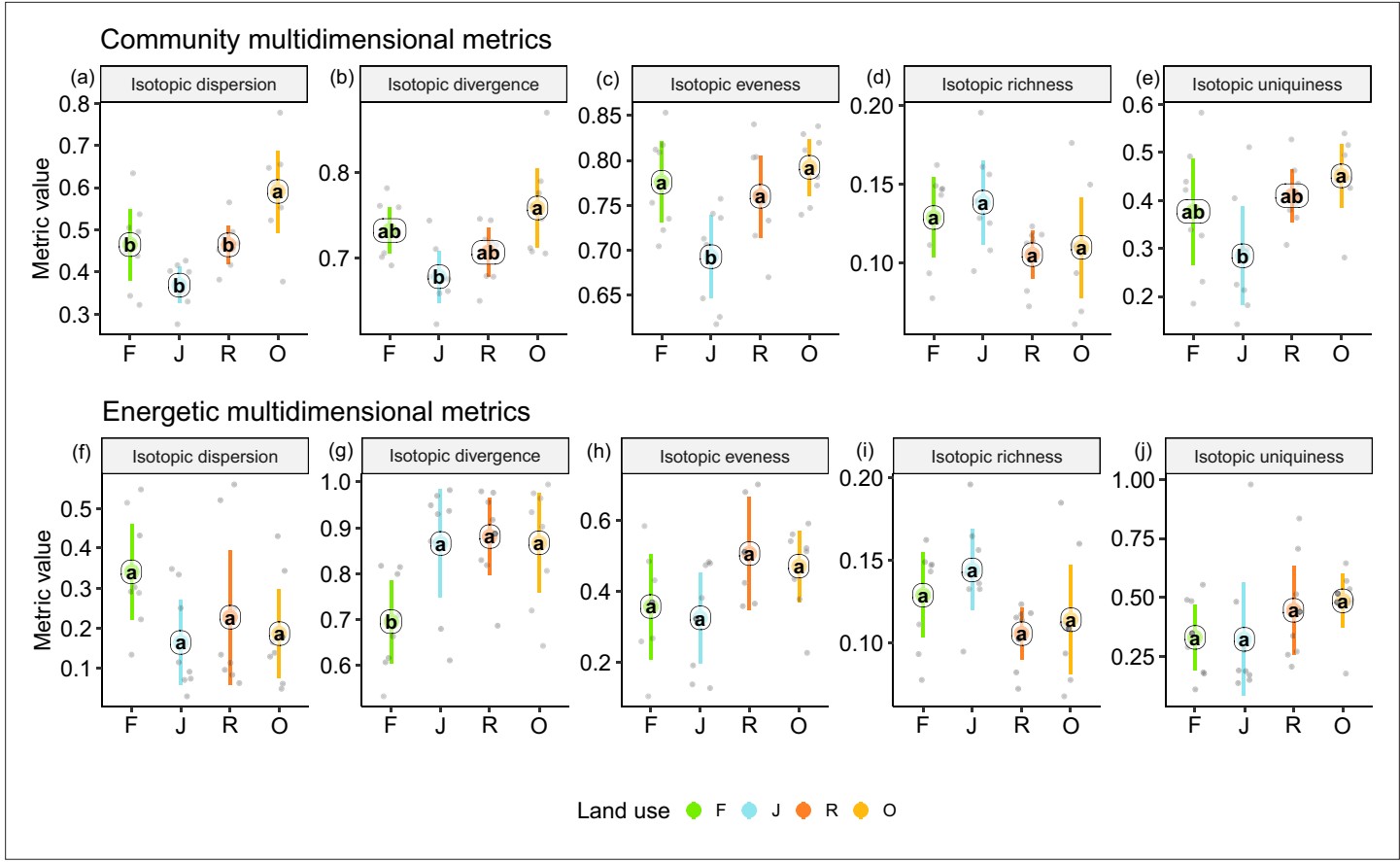

**Figure 4.** Multidimensional isotopic metrics of soil animal communities in rainforest (F, green), jungle rubber (J, blue), rubber (R, red), and oil palm plantations (O, yellow). Community (upper panel) and energetic metrics (lower panel) are shown. Each point represents one community (n = 8 per land-use system). Means sharing the same letter within each pane are not significantly different (Tukey's HSD test following ANOVA, p < 0.05).

The online version of this article includes the following source data and figure supplement(s) for figure 4:

**Source data 1.** Community metrics of soil food webs in each plot.

**Source data 2.** Energetic metrics of soil food webs in each plot.

**Figure supplement 1.** Multidimensional community metrics of soil food webs in forest (plot BF1).

**Figure supplement 2.** Multidimensional community metrics of soil food webs in forest (plot BF2).

**Figure supplement 3.** Multidimensional community metrics of soil food webs in forest (plot BF3).

**Figure supplement 4.** Multidimensional community metrics of soil food webs in forest (plot BF4).

**Figure supplement 5.** Multidimensional community metrics of soil food webs in jungle rubber (plot BJ2).

**Figure supplement 6.** Multidimensional community metrics of soil food webs in jungle rubber (plot BJ3).

**Figure supplement 7.** Multidimensional community metrics of soil food webs in jungle rubber (plot BJ4).

**Figure supplement 8.** Multidimensional community metrics of soil food webs in jungle rubber (plot BJ5).

**Figure supplement 9.** Multidimensional community metrics of soil food webs in oil palm plantation (plot BO2).

**Figure supplement 10.** Multidimensional community metrics of soil food webs in oil palm plantation (plot BO3).

**Figure supplement 11.** Multidimensional community metrics of soil food webs in oil palm plantation (plot BO4).

**Figure supplement 12.** Multidimensional community metrics of soil food webs in oil palm plantation (plot BO5).

**Figure supplement 13.** Multidimensional community metrics of soil food webs in rubber plantation (plot BR1).

**Figure supplement 14.** Multidimensional community metrics of soil food webs in rubber plantation (plot BR2).

**Figure supplement 15.** Multidimensional community metrics of soil food webs in rubber plantation (plot BR3).

**Figure supplement 16.** Multidimensional community metrics of soil food webs in rubber plantation (plot BR4).

**Figure supplement 17.** Multidimensional community metrics of soil food webs in forest (plot HF1).

*Figure 4 continued*

**Figure supplement 18.** Multidimensional community metrics of soil food webs in forest (plot HF2).

**Figure supplement 19** Multidimensional community metrics of soil food webs in forest (plot HF3).

**Figure supplement 20.** Multidimensional community metrics of soil food webs in forest (plot HF4).

**Figure supplement 21.** Multidimensional community metrics of soil food webs in jungle rubber (plot HJ1).

**Figure supplement 22.** Multidimensional community metrics of soil food webs in jungle rubber (plot HJ2).

**Figure supplement 23.** Multidimensional community metrics of soil food webs in jungle rubber (plot HJ3).

**Figure supplement 24.** Multidimensional community metrics of soil food webs in jungle rubber (plot HJ4).

**Figure supplement 25.** Multidimensional community metrics of soil food webs in oil palm plantation (plot HO1).

**Figure supplement 26.** Multidimensional community metrics of soil food webs in oil palm plantation (plot HO2).

**Figure supplement 27.** Multidimensional community metrics of soil food webs in oil palm plantation (plot HO3).

**Figure supplement 28.** Multidimensional community metrics of soil food webs in oil palm plantation (plot HO4).

**Figure supplement 29.** Multidimensional community metrics of soil food webs in rubber plantation (plot HR1).

**Figure supplement 30.** Multidimensional community metrics of soil food webs in rubber plantation (plot HR2).

**Figure supplement 31.** Multidimensional community metrics of soil food webs in rubber plantation (plot HR3).

**Figure supplement 32.** Multidimensional community metrics of soil food webs in rubber plantation (plot HR4).

**Figure supplement 33.** Multidimensional energetic metrics of soil food webs in forest (plot BF1).

**Figure supplement 34.** Multidimensional energetic metrics of soil food webs in forest (plot BF2).

**Figure supplement 35.** Multidimensional energetic metrics of soil food webs in forest (plot BF3).

**Figure supplement 36.** Multidimensional energetic metrics of soil food webs in forest (plot BF4).

**Figure supplement 37.** Multidimensional energetic metrics of soil food webs in jungle rubber (plot BJ2).

**Figure supplement 38.** Multidimensional energetic metrics of soil food webs in jungle rubber (plot BJ3).

**Figure supplement 39.** Multidimensional energetic metrics of soil food webs in jungle rubber (plot BJ4).

**Figure supplement 40.** Multidimensional energetic metrics of soil food webs in jungle rubber (plot BJ5).

**Figure supplement 41.** Multidimensional energetic metrics of soil food webs in oil palm plantation (plot BO2).

**Figure supplement 42.** Multidimensional energetic metrics of soil food webs in oil palm plantation (plot BO3).

**Figure supplement 43.** Multidimensional energetic metrics of soil food webs in oil palm plantation (plot BO4).

**Figure supplement 44.** Multidimensional energetic metrics of soil food webs in oil palm plantation (plot BO5).

**Figure supplement 45.** Multidimensional energetic metrics of soil food webs in rubber plantation (plot BR1).

**Figure supplement 46.** Multidimensional energetic metrics of soil food webs in rubber plantation (plot BR2).

**Figure supplement 47.** Multidimensional energetic metrics of soil food webs in rubber plantation (plot BR3).

**Figure supplement 48.** Multidimensional energetic metrics of soil food webs in rubber plantation (plot BR4).

**Figure supplement 49.** Multidimensional energetic metrics of soil food webs in forest (plot HF1).

**Figure supplement 50.** Multidimensional energetic metrics of soil food webs in forest (plot HF2).

**Figure supplement 51.** Multidimensional energetic metrics of soil food webs in forest (plot HF3).

**Figure supplement 52.** Multidimensional energetic metrics of soil food webs in forest (plot HF4).

**Figure supplement 53.** Multidimensional energetic metrics of soil food webs in jungle rubber (plot HJ1).

**Figure supplement 54.** Multidimensional energetic metrics of soil food webs in jungle rubber (plot HJ2).

**Figure supplement 55.** Multidimensional energetic metrics of soil food webs in jungle rubber (plot HJ3).

**Figure supplement 56.** Multidimensional energetic metrics of soil food webs in jungle rubber (plot HJ4).

**Figure supplement 57.** Multidimensional energetic metrics of soil food webs in oil palm plantation (plot HO1).

**Figure supplement 58.** Multidimensional energetic metrics of soil food webs in oil palm plantation (plot HO2).

**Figure supplement 59.** Multidimensional energetic metrics of soil food webs in oil palm plantation (plot HO3).

**Figure supplement 60.** Multidimensional energetic metrics of soil food webs in oil palm plantation (plot HO4).

**Figure supplement 61.** Multidimensional energetic metrics of soil food webs in rubber plantation (plot HR1).

**Figure supplement 62.** Multidimensional energetic metrics of soil food webs in rubber plantation (plot HR2).

**Figure supplement 63.** Multidimensional energetic metrics of soil food webs in rubber plantation (plot HR3).

**Figure supplement 64.** Multidimensional energetic metrics of soil food webs in rubber plantation (plot HR4).

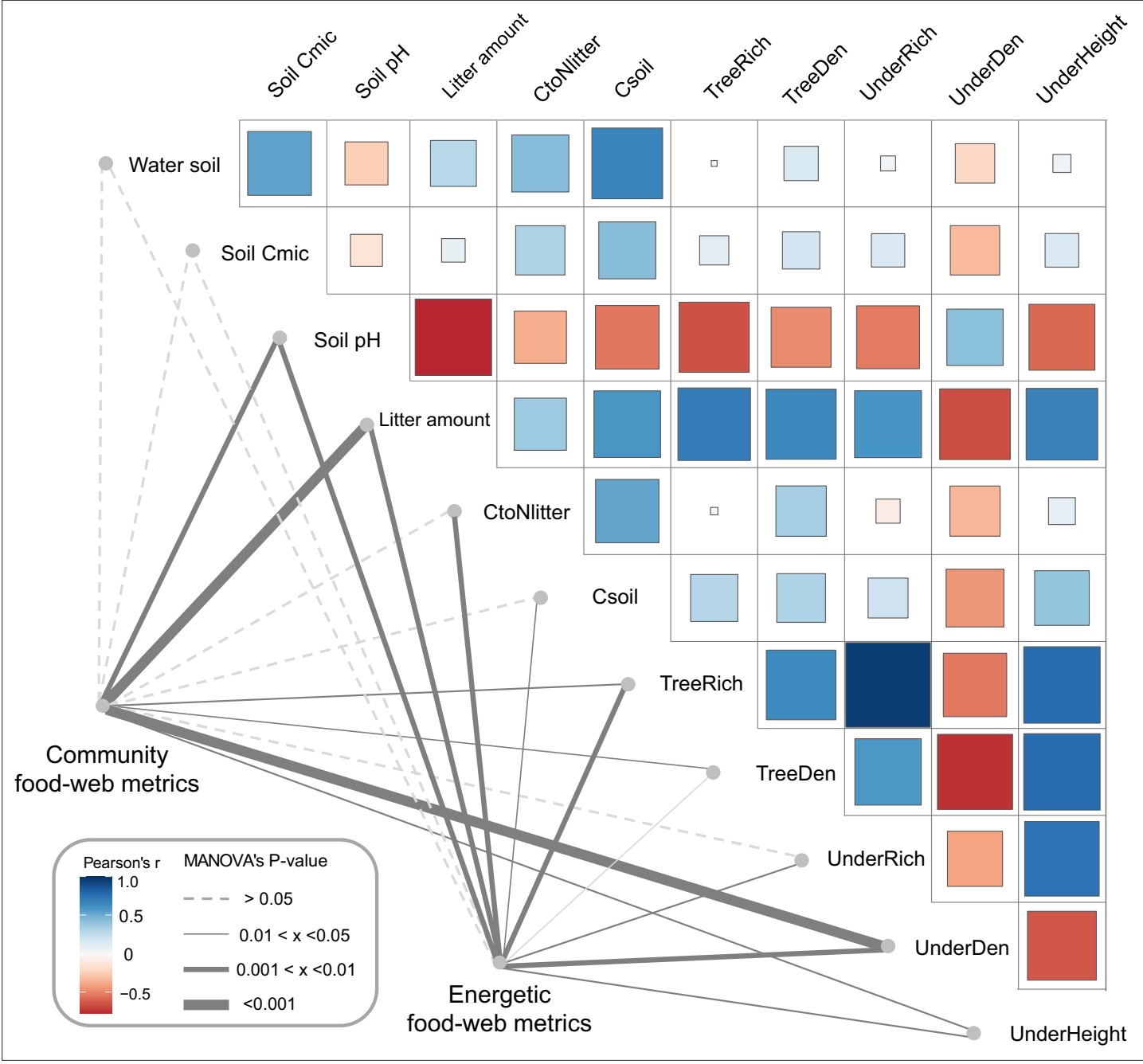

**Figure 5.** Environmental drivers of community and energetic soil food-web metrics. Community and energetic food-web metrics were related to environmental factors using multivariate analysis of variance (MANOVA); the thickness of connection lines shows statistical significance, dashed line for p > 0.05. Pairwise Spearman's correlations among environmental factors are shown with a tile chart (blue – negative, red – positive). The vegetation parameters included tree species richness (TreeRich), tree density (TreeDen), understory species richness (UnderRich), understory density (UnderDen), and average understory height (UnderHeight). Parameters of litter and soil include soil pH, litter amount, soil carbon concentration (Csoil), carbon-to-nitrogen ratio of litter (CtoNlitter), soil microbial biomass C (Soil Cmic), and soil humidity (Water soil) (***Krashevska et al., 2015***; ***Rembold et al., 2017a***).

The online version of this article includes the following source data for figure 5:

**Source data 1.** Data of environmental factors.

diversity metrics to land-use change were more pronounced for community than for energetic metrics. In agreement to our first hypothesis, [13]C values of animal taxa and communities were more enriched in rainforest than in plantations, but this shift vanished if the average $\Delta$[13]C values were weighted by metabolism. Soil animals in jungle rubber had the largest range of $\Delta$[15]N values among all land-use

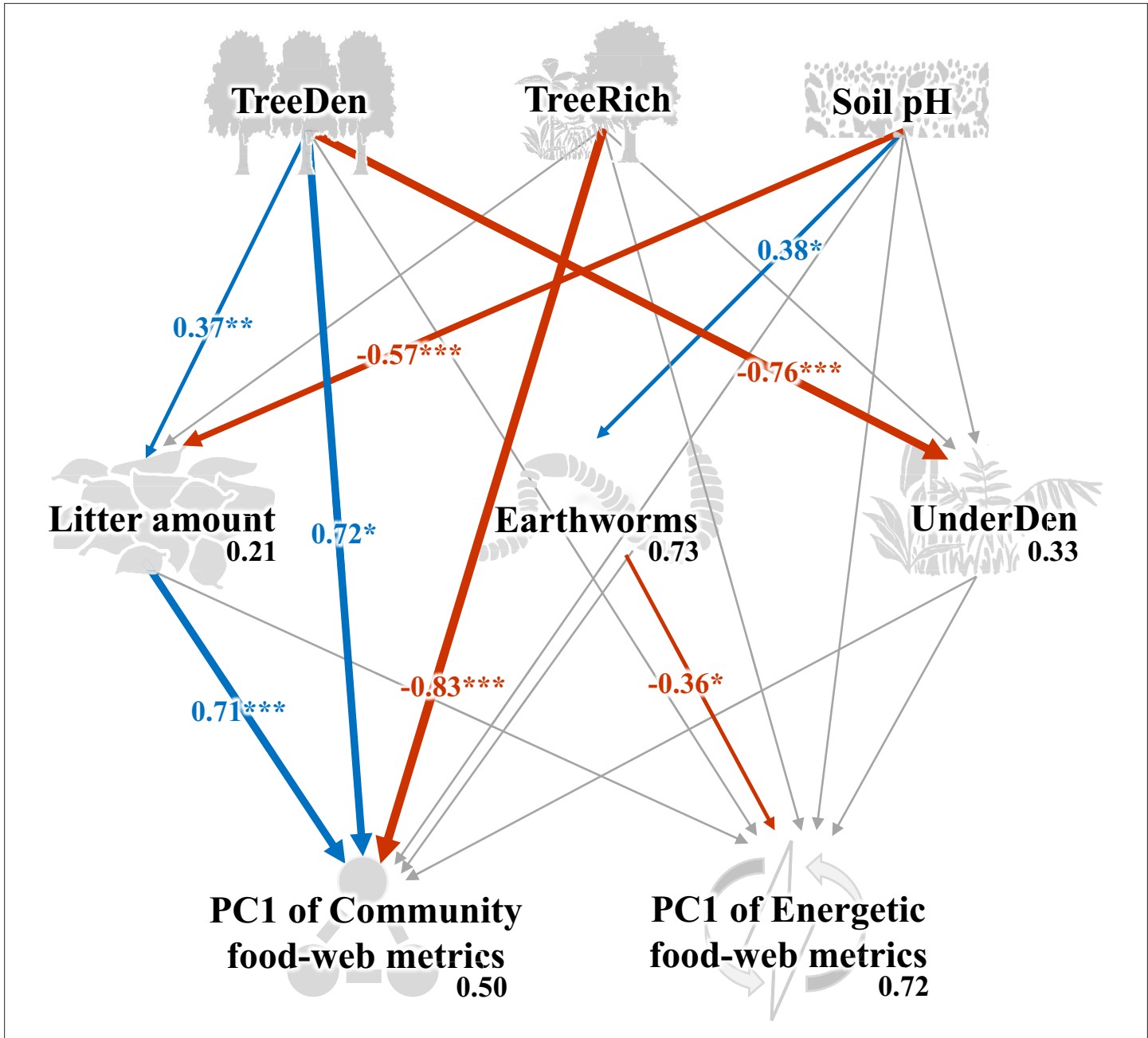

**Figure 6.** Structural equation model on the effects of environmental change on food-web metrics. Numbers adjacent to arrows are standardized path coefficients that show effect sizes and directions (blue – positive, red – negative) of the relationship, arrow width is proportional to the strength of path coefficients. Gray arrows represent paths that were not significant; *p < 0.05, **p < 0.01, and ***p < 0.001. Numbers above every response variable in the model denotes the proportion of variance explained. For abbreviations, see *Figure 5*.

The online version of this article includes the following source data for figure 6:

**Source data 1.** Data for building structural equation model (SEM).

systems, which suggests the longest food chain in this system. Refuting our second hypothesis, when considering all taxonomic groups being equally important ('community perspective'), soil food webs in oil palm had a significantly higher community dispersion than in the other land-use systems, and in trend also had a higher community divergence and uniqueness (i.e., a lower redundancy). By contrast, most energetic isotopic metrics ('energetic perspective') varied less between the four land-use systems. Conform to our third hypothesis, community isotopic metrics were more sensitive to changes in land use than energetic isotopic metrics, suggesting soil food webs are more sensitive to land-use

change from a community than from an energetic perspective. Further, community metrics of soil food webs were influenced directly by land use (tree properties and litter amount), whereas energetic metrics were influenced indirectly via pH-induced changes in earthworm abundance.

## The structure of tropical soil food webs

Our study is among the first comprehensive assessment of tropical soil food webs based on stable isotope analysis. Collembola, Symphyla, and Isopoda showed a much higher $^{15}$N enrichment than, for example, Oribatida, but all three groups occupy similar trophic positions in temperate forests and predominantly function as decomposers (*Potapov et al., 2019c*). This difference may be caused by low litter quality in tropical forests forcing decomposers to switch to more microbial or even animal diet (*Illig et al., 2005*). Protura in temperate forests are enriched in $^{15}$N and feed on ectomycorrhizal fungi (*Bluhm et al., 2019*), whereas in the studied tropical forests, Protura had the lowest $\Delta^{15}$N values among all groups, suggesting that they feed on saprotrophic rather than mycorrhizal fungi. The low $\Delta^{15}$N and high $\Delta^{13}$C values of Pauropoda, reported for the first time for this group, indicate that they function as decomposers by feeding on saprotrophic microorganisms (*Tiunov et al., 2015*), confirming earlier suggestions (*Starling, 1944*). Low $\Delta^{15}$N values in Protura, Diplopoda, Isoptera, Psocoptera, and Blattodea may be associated with feeding on algae (*Potapov et al., 2018*), shown to be important food for mesofauna in tropical soil food webs (*Susanti et al., 2019*; *Semenina et al., 2020*). Unexpectedly, micropredators (e.g., Diplura and Pseudoscorpiones) had higher trophic positions ($\Delta^{15}$N values) than macropredators (e.g., Araneae and Formicidae) across all land-use systems, and Diplura had the highest $\Delta^{15}$N values among all taxa studied. Diplura were represented mostly by predatory Japygidae, which may hunt springtails, mites, and other small invertebrates (*Sendra et al., 2021*). The higher trophic position of small-sized predators suggests that they form part of a different energy channel than macropredators. In fact, the micro-food web in soil has been shown to be based mainly on microbial resources channeled to higher trophic levels by microarthropod predators, whereas the macro-food web is based more on litter and detritus consumed by macrofauna taxa with the energy channeled to higher trophic levels by macroarthropod predators (*Potapov et al., 2021a*). This implies more trophic transactions in the micro-food web (*Pollierer et al., 2009*; *Steffan et al., 2015*) explaining the higher trophic position of micro- than macroarthropod predators. Among other groups with a wide food spectrum, Diptera had higher isotopic values (both $\Delta^{15}$N and $\Delta^{13}$C) than Coleoptera, indicating that flies are more closely linked to detrital and microbial food chains in tropical soil food webs than beetles. Isopoda had strikingly high $\Delta^{15}$N values for macrodecomposers, possibly due to intense coprophagy (*Potapov et al., 2022*). Overall, despite general similarities, we also found consistent differences between tropical and temperate soil food webs. Further studies comparing differences in soil food-web structure and associated soil functions across temperate and tropical ecosystems are needed to prove the generality of these differences and their consequences for biodiversity – ecosystem functioning relationships.

## Shifts in trophic positions of soil invertebrates and energy channels in soil food webs

In agreement with our first hypothesis, $\Delta^{13}$C values of most of the studied soil animal taxa were higher in rainforest than in plantations. The $^{13}$C concentration in dead plant material is increasing during decomposition compared to fresh leaf litter (*Ågren et al., 1996*; *Boström et al., 2007*; *Potapov et al., 2019c*), and high $\Delta^{13}$C values in soil fauna in forest likely indicate feeding on saprotrophic fungi and bacteria that assimilate predominantly labile $^{13}$C-enriched plant compounds (*Pollierer et al., 2009*; *Potapov et al., 2013*; *Hyodo, 2015*). Vascular and non-vascular plants have generally lower $\Delta^{13}$C values than saprotrophic microorganisms and animals (*Hyodo et al., 2010*; *Potapov et al., 2019c*), therefore, the high $\Delta^{13}$C values in soil invertebrates in rainforest point to a more pronounced detritus-based 'brown' food web relying heavily on saprotrophic fungi and bacteria based on litter material. Among the plantations, the unweighted average $\Delta^{13}$C values were lowest in oil palm suggesting a shift toward a more plant-based 'green' food web relying more heavily on the consumption of living plant tissue (*Fujii et al., 2021*), which has been previously shown for Chilopoda, Oribatida, Collembola, and Pseudoscorpiones at the same study sites (*Klarner et al., 2017*; *Krause et al., 2019*; *Liebke et al., 2021*; *Susanti et al., 2021*). Results of the study of *Susanti et al., 2019*, using fatty acids as trophic biomarkers at our study sites further support the conclusion of a more pronounced plant- and reduced

detritus-based energy channel in soil food webs of plantations compared to rainforest. Compared to rainforest the herb layer is much more developed in plantations due to more open canopy and coverage by weeds (*Rembold et al., 2017a*), presumably providing high-quality resources for plant and litter feeding soil animals. By contrast, litter in rainforest is high in lignin and therefore of low food quality (*Krashevska et al., 2018*), increasing the use of saprotrophic microorganisms rather than litter by detritivores (*Illig et al., 2005*).

In soil food-web models the bacterial energy channel typically is considered to be 'fast', while the fungal energy channel is considered to be 'slow' due to differences in the turnover rates of bacteria and their consumers, and fungi and their consumers, respectively (*Coleman et al., 1983*; *Moore et al., 2005*). Generally, the fast energy channel is defined by fast-growing populations with short turnover rates (*Rooney et al., 2006*). Based on this concept, animals feeding on living plants or fresh litter contribute to fast energy channeling, whereas animals feeding on recalcitrant detritus (being slowly decomposed belowground) are associated with slow energy channeling. This perspective complements the green (plant) and brown (detrital) energy channels in soil food webs. Since $^{13}$C accumulates in plant-derived materials with microbial decomposition (*Pollierer et al., 2009*; *Potapov et al., 2013*; *Fujii et al., 2021*), $\Delta^{13}$C values in soil animals may indicate slow versus fast carbon cycling in soil. Therefore, the shift toward plant-based energy channeling in food webs of plantations may reflect accelerated carbon cycling at the ecosystem level, contributing to carbon losses of plantations (*Guillaume et al., 2018*) and thereby potentially compromising ecosystem long-term stability (*McCann et al., 1998*; *Rooney and McCann, 2012*).

Contrasting the community perspective, energetic $\Delta^{13}$C values of communities did not vary significantly among land-use systems, which was due to similar $\Delta^{13}$C values of Annelida (earthworms) and some other macro-decomposers across land-use systems. Earthworms had the highest share in community metabolism among detritivores in plantations suggesting that they predominate animal-mediated decomposition and organic matter transformation processes (*Potapov et al., 2019a*). Notably, earthworms were among the most $^{13}$C-enriched animal groups in jungle rubber, rubber and oil palm plantations, but their $\Delta^{13}$C values were similar to other soil animal groups in rainforest. Earthworms feed on detritus and microorganisms and are able to efficiently use fresh litter carbon, but also 'old' microbially processed carbon (e.g., soil organic matter; *Scheu and Falca, 2000*; *Hyodo et al., 2012*; *Blouin et al., 2013*) reflected in high $\Delta^{13}$C values (*Pollierer et al., 2009*). Land-use change in tropical lowland landscapes is associated with the loss of biodiversity and reduced biomass of litter arthropods (*Barnes et al., 2014*), but the negative effect of biodiversity loss on soil functions may be at least in part counteracted by earthworms that monopolize the detrital channel in plantations. Thereby, earthworms also may counteract the destabilization of the system through sequestration of carbon in their large body and thus strengthening 'slow' energy channeling (*Rooney and McCann, 2012*; *Schwarzmüller et al., 2015*). Earthworms contribute only about 15.4% to community metabolism in rainforest, leaving vacated trophic space for other groups that vanish or are reduced in plantations. Combined, high $\Delta^{13}$C values and shift in dominance of detritivore taxa suggest that the detrital energy channel in rainforest is diversified and comprises a wider range of consumer groups than in plantations, whereas in plantations it comprises almost exclusively Annelida. The similar weighted average $\Delta^{13}$C values in plantations and rainforest suggest that from an energetic perspective, soil food webs in plantations are as efficient in processing old organic carbon as in rainforest, despite having a very different structure. However, the decomposer system in plantations may be more vulnerable against future changes since their functioning relies on a single detritivore group comprising predominantly a single invasive species (*Potapov et al., 2021b*).

Unlike $\Delta^{13}$C, changes in $\Delta^{15}$N with changes in land use were less consistent across animal groups. Lower $\Delta^{15}$N values in Oribatida and Blattodea in rainforest compared to plantations may be linked to their trophic plasticity, allowing them to shift their trophic position from primary decomposer in rainforest to secondary decomposer in litter-poor plantations (*Krause et al., 2019*). By contrast, many taxa had lower trophic positions in oil palm compared to rainforest, including Hemiptera, Orthoptera, Isoptera, Pseudoscorpiones, Diplopoda, and Psocoptera. The low $\Delta^{15}$N values in these taxa at least in part may be due to feeding on resources depleted in $\Delta^{15}$N relative to litter, such as algae and lichens (*Chahartaghi et al., 2005*; *Schneider et al., 2004*), suggesting that non-vascular plants may play a more important role for groups such as Hemiptera, Orthoptera, and Isoptera in plantations than in rainforest, potentially associated with the more open canopies in plantations.

The range of Δ¹⁵N values in soil communities reflects the length of food chains (*Cabana and Rasmussen, 1994*; *Scheu and Falca, 2000*), and was the largest in jungle rubber. This was caused by the very low Δ¹⁵N values of Pauropoda (–5.1‰) and Orthoptera (–2.1‰) and high Δ¹⁵N values of Diplura (10.0‰). Jungle rubber is a system that is highly heterogeneous in management practices and plant richness (*Gouyon et al., 1993*; *Rembold et al., 2017b*), with species richness in some arthropod predators even exceeding that in rainforest at our study sites (*Junggebauer et al., 2021*). Anthropogenic disturbances in jungle rubber are moderate compared to monoculture plantation systems (*Barnes et al., 2014*) and food chains have been found to be longest at intermediate levels of disturbance (*Menge and Sutherland, 1987*; *Polis and Winemiller, 2018*; *Post, 2002b*), which may explain the largest range of Δ¹⁵N values in jungle rubber. As a note of caution, however, the δ¹⁵N values of primary producers (vascular plants, algae, and lichens) may vary among our study systems, which may have affected the Δ¹⁵N values of consumers, but unlikely our overall conclusions.

## Changes in functional diversity of soil food webs from community and energetic perspectives

Refuting our second hypothesis, neither isotopic diversity nor isotopic redundancy were higher in rainforest than in plantations. However, isotopic richness was slightly higher in the two more natural systems (i.e., rainforest and jungle rubber) than in rubber and oil palm plantations. Oil palm showed significantly higher community dispersion values than the other land-use systems and in trend had the highest community divergence (unweighted values) reflecting the proportion of groups with the most extreme trophic (isotopic) niches within the community (*Cucherousset and Villéger, 2015*; *Mason et al., 2005*; *Villéger et al., 2008*). At least in part this likely was due to feeding on non-vascular plants, such as algae and lichens, characterized by very different stable isotope values than C3 plants, that is, the dominant vegetation at our study sites (*Potapov et al., 2019c*). As discussed above, the more open canopy in plantations favors algae and lichens (*Drescher et al., 2016*; *Schulz et al., 2019*), together with the monopolization of detrital channel by earthworms (representing another 'extreme' isotopic niche), the use of non-vascular plants explains the high dispersion and divergence of energy channeling in oil palm plantations.

From the 'energetic perspective', soil food-web divergence in plantations was significantly higher than in rainforest. This contrasts previous evidence that functional divergence decreases with disturbance (*Gerisch et al., 2012*; *Mouillot et al., 2013*). However, contrary to divergence, energetic dispersion was in trend higher in rainforest than in the other land-use systems. Similar to the community metrics, the energetic metrics indicated that food-web characteristics in plantations deviate from those in rainforest (high divergence), with food webs being less balanced (low dispersion) with most of the energy being channeled and locked into earthworms.

Community isotopic uniqueness, defined as the inverse of the average isotopic redundancy, and community evenness (*Cucherousset and Villéger, 2015*) were low in jungle rubber. Most of the soil animal groups clustered in a small region in stable isotope space in jungle rubber, resulting in low community uniqueness, while Pauropoda and Orthoptera were far from this cluster, resulting in low community evenness. High functional redundancy may buffer against land-use impacts and promote stable food webs with long trophic chains (*Brodie et al., 2014*; *Chua et al., 2021*; *Sanders et al., 2018*), which is supported by the results of our study. The increase in isotopic uniqueness (both energetic and unweighted) in oil palm plantations may reflect eroded resilience of this system against future changes.

Overall, community food-web metrics were more sensitive to changes in land use than energetic food-web metrics, suggesting that compositional changes in soil food webs with land use are stronger than changes in energy channeling. This echoes earlier findings that land-use effects on soil animal biodiversity exceed those on functional diversity (*Potapov et al., 2020*). Results of our SEM indicated significant direct effects of land use-induced environmental changes (i.e., litter amount, tree density, and tree species richness) on community food-web metrics. The amount and quality of leaf litter are important drivers of soil fauna composition and soil food-web structure, being both the food and the habitat for soil animals (*Fujii et al., 2020*; *Sayer et al., 2006*). Apart from litter-mediated effects, changes in tree density and species richness are associated with changes in root-derived resources (*Ballauff et al., 2021*), which also fuel belowground food webs (*Pollierer et al., 2007*; *Bradford, 2016*), and this may explain the direct effects of tree communities on the food-web metrics. By contrast,

energetic food-web metrics were not directly affected by changes in tree communities and pH, but were linked to the changes in earthworm abundance. High soil pH favors colonization of plantations by earthworms and this is common in the tropics (*Marichal et al., 2010*; *Potapov et al., 2021b*). The close association between energetic food-web metrics and the fraction earthworms contribute to community metabolism stems to a large extent from the mathematical dependence between these two variables. However, we intentionally wanted to illustrate that those strong shifts in the functional diversity of food webs may result from a single group benefiting from certain environmental changes.

In conclusion, our study is among the first comprehensive assessment of tropical soil food webs and their variation due to land-use changes. Low $\Delta^{13}C$ values in most soil animal groups in plantations in comparison to rainforest indicate a shift toward using plant carbon and 'fast' energy channeling based on high-quality understory plants (weeds) as well as algae. On the other hand, the trophic niche of earthworms as major macrofauna detritivores stayed unchanged and they monopolized the 'slow' detrital channel in plantations. This resulted in systems with strong divergence and imbalance in energetic pathways potentially compromising functional stability of plantation systems. Other studies at our sites showed that these changes in soil food-web characteristics with transformation of rainforest into plantations are associated with reduced soil functioning (*Grass et al., 2020*) and litter invertebrate biodiversity (*Barnes et al., 2014*). Our analyses allowed to uncover the mechanisms responsible for these changes and demonstrated that land-use effects on soil biodiversity from a 'community perspective' are in part buffered from the perspective of energy channeling ('energetic perspective'), but resistance of plantations against future changes in climate and land use may be compromised.

## Materials and methods
### Sampling sites
The study was conducted in the framework of the collaborative research project CRC990/EFForTS investigating ecological and socio-economic changes associated with the transformation of lowland rainforest into agricultural systems (*Drescher et al., 2016*). Four land-use systems, rainforest, jungle rubber, rubber plantations, and oil palm plantations were investigated in two regions, that is, Harapan and Bukit Duabelas (*Drescher et al., 2016*). Jungle rubber sites were established by planting rubber trees (*Hevea brasiliensis*) into selectively logged rainforest and contain rainforest tree species. Jungle rubber sites represent low intensive land-use systems, lacking fertilizer input as well as herbicide application; the age of rubber trees varied between 15 and 40 years (*Kotowska et al., 2015*). Rubber and oil palm (*Elaeis guineensis*) monocultures represent high land-use intensity plantation systems managed by the addition of fertilizers as well as herbicides (*Drescher et al., 2016*). Each land-use system was replicated four times in each landscape, resulting in a total of 32 sites; for more details, see *Drescher et al., 2016*.

### Sampling, extraction, and classification of soil fauna
Soil animals were sampled at each of 32 study sites during October and November 2013. Soil samples measuring 16 cm × 16 cm and including the litter layer and 0–5 cm of the mineral soil were taken in three 5 m × 5 m subplots within each of 50 m × 50 m plots established at each study site, resulting in a total of 96 samples. The samples were transported to the laboratory and animals were extracted by heat (*Kempson et al., 1963*) until the substrate was completely dry (6–8 days). Until further analysis, species were stored in 70% ethanol. For calibration of the animal stable isotope values, we used mixed litter samples that were taken from each site and analyzed in a previous study (*Klarner et al., 2017*).

Animals were classified into 23 high-rank taxonomic groups (Oribatida, Collembola, Symphyla, Protura, Annelida, Blattodea, Diplopoda, Isopoda, Isoptera, Psocoptera, Psocoptera, Lepidoptera, Orthoptera, Thysanoptera, Diptera, Coleoptera, Pseudoscorpiones, Mesostigmata, Diplura, Formicidae, Chilopoda, Araneae, Pauropoda). For stable isotope analysis, we adopted a group-level analysis representing the stable isotope niche at the level of taxonomic groups. Although this approach may miss the variability in stable isotope niches of species within high-rank taxonomic groups, it has the advantage that it integrates across species allowing generalizations on the trophic structure and energy flux of whole communities. The approach has been recently advocated for analyzing the channeling of energy through food webs using lipid profiling (*Kühn et al., 2018*), but has not been adopted yet in stable isotope analysis although it has been shown that at least in soil high-rank animal

taxa typically represent the trophic niches of species in most taxa (**Potapov et al., 2019b**). To ensure that our samples reliably represent the trophic niche of the studied taxa, we included (if ever possible) 15 individuals per taxon per study site. Doing that we considered the turnover of species among sites and focused on dominant species representing the majority of biomass, which we considered most important for our approach. We further classified taxonomic groups into five major functional groups according to their trophic guild and body size class (**Potapov et al., 2019b**; **Potapov et al., 2021a**): herbivores including, for example, Hemiptera and Orthoptera, microdecomposers including, for example, Oribatida and Collembola, macrodecomposers including, for example, Annelida and Diplopoda, micropredators including, for example, Diplura and Mesotigmata, macropredators including, for example, Araneae and Chilopoda, and groups with mixed feeding habits including, for example, Diptera and Coleoptera.

## Stable isotope analysis

To cover the entire community, for each sampling site we analyzed all taxa for which we were able to collect enough biomass for stable isotope analysis and which were represented by more than two individuals. We analyzed a minimum of 3 and a maximum of 15 individuals for each taxonomic group for each site as a single mixed sample to cover the species- and individual-level isotopic variation. We mixed individuals from different subplots whenever possible to cover spatial variation in stable isotope values. Animals from the litter and soil layer were analyzed separately, but were merged for data analysis since stable isotope values did not differ significantly between layers. Animal samples were dried at 60°C for 24 hr before stable isotope analysis, weighed and wrapped into tin capsules; sample weight varied between 0.01 and 1.00 mg. For small-sized animal groups we used bulk individuals, for large-sized animal groups we used body parts dominated by muscle tissue (e.g., legs) from different individuals and pooled them (**Tsurikov et al., 2015**). In total, 626 samples of 23 taxonomic groups were analyzed across 32 sites. For Collembola, Oribatida, and Chilopoda, we additionally used stable isotope data collected at species level (**Klarner et al., 2017**; **Krause et al., 2019**; **Susanti et al., 2021**) to calculate a single average value for each group at each site, which were collected at the same sampling campaign. The number of analyzed taxonomic groups varied between 6 and 17 per site (i.e., per one soil food web) and was on average 12.3.

Animal samples were analyzed using a coupled system of an elemental analyzer (NA 1500, Carlo Erba, Milan, Italy) and a mass spectrometer (MAT 251, Finnigan, Bremen, Germany) adopted for the analysis of small sample sizes (**Langel and Dyckmans, 2014**). Ratios of the heavy isotope to the light isotope ($^{13}$C/$^{12}$C,$^{15}$N/$^{14}$N, denoted as R) were expressed in parts per thousand relative to the standard using the delta notation with $\delta^{13}$C or $\delta^{15}$N = ($R_{sample}$/$R_{standard}$ − 1) × 1000 (‰). Vienna PD Belemnite and atmospheric nitrogen were used as standard for $^{13}$C and $^{15}$N, respectively. Acetanilid was used for internal calibration.

Environmental parameters of the study sites were used as given in **Potapov et al., 2020**, **Krashevska et al., 2015**, and **Rembold et al., 2017a**, which included tree species richness, tree density, understory species richness, understory density, and average understory height, soil pH, litter amount, soil carbon concentration, carbon-to-nitrogen ratio of litter, soil microbial biomass C, and soil humidity.

## Statistical analysis

The stable isotope compositions of animals were calibrated to that of the local leaf litter. Calibrated $\delta^{13}$C and $\delta^{15}$N values were calculated as the difference between the plot-specific litter $\delta^{13}$C and $\delta^{15}$N values and the $\delta^{13}$C and $\delta^{15}$N values of each group, and given as $\Delta^{13}$C and $\Delta^{15}$N values, respectively. Statistical analyses were done in R v 4.0 (**R Development Core Team, 2020**) with R studio interface (**RStudio Team, 2020**).

To characterize the trophic structure of soil animal communities, we calculated isotopic metrics as given in **Cucherousset and Villéger, 2015**. One-dimensional metrics describe the isotopic parameters of the communities based on $\Delta^{15}$N or $\Delta^{13}$C values. Multidimensional metrics combine both $\Delta^{13}$C and $\Delta^{15}$N values, and join the ones from **Layman et al., 2007**, with functional diversity framework (**Villéger et al., 2008**; **Laliberté and Legendre, 2010**). The $\Delta^{13}$C and $\Delta^{15}$N values were scaled between 0 and 1 based on maximum and minimum across all communities to ensure equal contribution of two isotopes prior to calculation of multidimensional metrics. Multidimensional metrics were calculated from two perspectives: (1) a 'community perspective', assuming all taxonomic groups being equally important,

that is, *unweighted metrics*, and (2) an 'energetic perspective', assuming that groups that have higher contribution to total community metabolism are also more functionally important, that is, metrics were *weighted by community metabolism*. We used metabolism instead of biomass because it better reflects the contribution of organisms to energy processing and thus their importance in the food web (*Brown et al., 2004*; *Barnes et al., 2018*). Community metabolism for each group at each plot was taken from *Potapov et al., 2019a*; it was based on length and width measurements of all individuals and using body size to body mass ratios and group-specific allometric regressions to calculate metabolic rates (*Ehnes et al., 2011*). Individual metabolic rates were then summed up for groups to estimate contribution of each taxonomic group to the total community metabolism per plot (*Supplementary file 1*, *Figure 1—figure supplement 1*).

Overall, 13 isotopic metrics were calculated for each of 32 communities (i.e., sampling plots). One-dimensional metrics included average position, range, minimum, and maximum. The unweighted and metabolism-weighted *average position* of communities (mean isotopic value across groups) represent mean community-level isotopic trait values. The *isotopic range* represents the difference between *minimum* and *maximum* values of both $\Delta^{13}C$ and $\Delta^{15}N$. Range, minimum, and maximum could not be weighted and are given unweighted. Multidimensional metrics included isotopic divergence, isotopic dispersion, isotopic evenness, isotopic uniqueness, and isotopic richness, which were calculated as both unweighted and metabolism-weighed. *Isotopic divergence* represents the distance between all species and the center of the convex hull area. Isotopic divergence values close to 0 indicate that groups with extreme stable isotope values are rare (community divergence) or contribute little to the community metabolism (energetic divergence), whereas isotopic divergence values close to 1 indicate that there are many groups with extreme stable isotope values (community divergence) or they contribute considerably to the community metabolism (energetic divergence). *Isotopic dispersion* combines convex hull area with isotopic divergence values and can be interpreted as scaled multidimensional variance. Isotopic dispersion approaches 1 when species with contrasting stable isotope values have similar abundance, which is a more functionally diverse and balanced system, whereas it approaches 0 when most groups (community dispersion) or community metabolism (energetic dispersion) are concentrated near the 'center of gravity' of the community in stable isotope space. *Isotopic evenness* quantifies the distribution of groups or metabolism in stable isotope space. Isotopic evenness values close to 1 indicate that the isotope values of the groups/metabolism are evenly distributed, while values close to 0 indicate that the groups/metabolism cluster together. *Isotopic uniqueness* reflects the closeness of stable isotope values of the studied groups/metabolism within the community, which is defined as the inverse of the average isotopic redundancy. Finally, *isotopic richness* is the volume occupied by all groups in isotopic space (convex hull area in two-dimensional isotopic space) and reflects functional richness of the food web; it is the only multidimensional metric that cannot be weighted since it considers the total isotopic space (*Mason et al., 2005*; *Villéger et al., 2008*).

To assess differences in food-web structure among land-use systems, we used a set of analyses of variance (*aov* function) with the $\Delta^{13}C$ and $\Delta^{15}N$ values of each taxonomic group, one-dimensional isotopic metrics, and multidimensional community and energetic isotopic metrics as response variables, and land-use system (rainforest, jungle rubber, rubber, oil palm) and landscape (Harapan or Bukit Duabelas) as factors (total n = 32, 8 plots as replicates per land-use system). Pairwise comparisons of means among land-use systems were done using post hoc *HSD.test* function from the package *agricolae Margur, 2020* following analyses of variance. Differences in $\Delta^{13}C$ and $\Delta^{15}N$ values between rainforest and other land-use systems for each taxonomic group were analyzed with Student's *t.test* function in R. Results were visualized using the *ggplot2* package (*Wickham, 2016*).

To assess effect size of land use on all food-web metrics combined, we used analysis of similarities based on community and energetic metrics with land use as the grouping variable (*anosim* in package *vegan*). Besides, we used multivariate analyses of variance (MANOVAs) to inspect the effects of environmental factors on community and energetic metrics, and additionally explored pairwise correlations between environmental factors and food-web metrics using Spearman's correlation from the package *agricolae* (*Margur, 2020*).

Finally, a SEM based on generalized least squares was constructed to provide insight into how land use affected soil food webs from both community and energetic perspectives. The analysis was performed with the *lavaan* package in R (*Rosseel, 2012*). The model included tree, understory, and

soil properties selected according to permutation tests based on R² which were used to quantify the land-use effects (*ordiR2step* in package *vegan*), and before permutation tests, the environmental factors were filtered based on the MANOVAs and Spearman's correlation. The final model included soil pH, tree density, and tree richness as the three most important variables that represented direct land-use effects (i.e., logging and liming; *Drescher et al., 2016*). Furthermore, we included litter amount, understory density, and earthworm metabolism as the three mediators that are affected by changes in tree density, tree richness, and pH, and have strong impacts on tropical soil invertebrate communities (*Darras et al., 2019*; *Potapov et al., 2019a*). Food-web metrics (i.e., isotopic divergence, dispersion, uniqueness, evenness, and average position) were first combined using principal component analysis (*prcomp* function) and the PC1 was used as the response variable in SEM; PC1 explained 66% and 42% of the variance for community and energetic metrics, respectively. To determine the goodness of fit of the model, we used $\chi^2$-test associated p-value $\geq 0.05$, the comparative fit index (CFI) > 0.95, the root-mean-square error of approximation (RMSEA) and the standardized root-mean-square residual (SRMR) with values $\leq 0.05$ (*Schermelleh-Engel et al., 2003*). Our SEM adequately described the data ($\chi^2 = 6.23$, p = 0.40, df = 5, CFI = 0.98, RMSEA = 0.04, SRMR = 0.05).

## Acknowledgements

This study was funded by the Deutsche Forschungsgemeinschaft (DFG), project number 192626868-SFB 990 in the framework of the collaborative German-Indonesian research project CRC990. ZZ are supported by China Scholarship Council (CSC) (202004910314). We thank Dr Katja Rembold and Prof Holger Kreft for providing vegetation parameters; we also thank Zhijing Xie and Haifeng Yin for discussion. Special gratitude goes to Svenja Meyer for the animal silhouettes. We acknowledge support by the Open Access Publication Funds of the University of Göttingen.

## Additional information

### Funding

| Funder | Grant reference number | Author |
|---|---|---|
| Deutsche Forschungsgemeinschaft | 192626868-SFB 990 | Valentyna Krashevska Rahayu Widyastuti Stefan Scheu Anton Potapov |
| China Scholarship Council | 202004910314 | Zheng Zhou |

The funders had no role in study design, data collection and interpretation, or the decision to submit the work for publication.

### Author contributions

Zheng Zhou, Data curation, Formal analysis, Visualization, Writing - original draft, Writing – review and editing; Valentyna Krashevska, Data curation, Writing – review and editing; Rahayu Widyastuti, Investigation; Stefan Scheu, Conceptualization, Funding acquisition, Supervision, Writing – review and editing; Anton Potapov, Conceptualization, Funding acquisition, Methodology, Supervision, Writing – review and editing

### Author ORCIDs

Zheng Zhou http://orcid.org/0000-0002-8078-6378
Valentyna Krashevska http://orcid.org/0000-0002-9765-5833
Stefan Scheu http://orcid.org/0000-0003-4350-9520
Anton Potapov http://orcid.org/0000-0002-4456-1710

### Decision letter and Author response

Decision letter https://doi.org/10.7554/eLife.75428.sa1
Author response https://doi.org/10.7554/eLife.75428.sa2

## Additional files

### Supplementary files

• Supplementary file 1. Metabolism proportion of different animal groups in different land-use systems.

• Transparent reporting form

### Data availability

All data generated or analysed during this study are included in the manuscript and supporting file.

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
