## [Editor Report]

Zhou et al., provide a robust study on isotopic and metabolic changes of a soil community across a gradient of different land-use types in Sumatra, Indonesia. By mixing community-based analyses of stable isotopes and size-based metabolic measures, they are able to elucidate, for the first time, important links among plants and the soil food web in tropical ecosystems. This study is of importance to tropical biologists, ecosystem ecologists and biodiversity conservationists aiming to understand the impacts of humans on tropical forests.

---

## [Decision Letter]

**Decision letter after peer review:**

Thank you for submitting your article "Tropical land use alters functional diversity of soil food webs and leads to monopolization of the detrital energy channel" for consideration by *eLife*. Your article has been reviewed by two peer reviewers, oe of whom is a member of our Board of Reviewing Editors, and the evaluation has been overseen by Christian Rutz as the Senior Editor. The reviewers have opted to remain anonymous.

The reviewers have discussed their reviews with one another, and the Reviewing Editor has drafted this decision letter to help you prepare a revised submission.

Essential revisions:

1) As one of the reviewers noted, the use of stable isotopes on higher taxonomic ranks needs to be fully re-evaluated and much better justified. Since these analyses are critical to supporting the study's claims, this point requires careful revisions.

2) Other comments and requests of expanding the discussion (likely in an appendix) require attention.

*Reviewer #1 (Recommendations for the authors):*

The Introduction is already long, but it may benefit from an expanded discussion of previous literature on metabolic rates and soil communities (beyond setting the fast-bacteria and slow-fungi framework). You get to know that this ms deals with metabolic rates of inverts just in the Objectives, and you get to know that metabolic data come from previous literature in M&M.

For me, one major shortcoming this manuscript has is the very reduced Discussion of specific taxa. I agree that expanding a Discussion to 23 invertebrate taxa may be too much. But, the manuscript does little to satisfy specific questions arising on specific taxa and thus its ability to generate testable hypothesis remains low. An expanded Discussion in an Appendix, maybe accompanying the graphs already in there, may be useful for this purpose. Just think about the many *single* taxa taxonomists that will read this ms and wish to read more about their preferred taxa. You can use such an appendix to set ideas to test in the future.

Please correct the many ways to refer to Mr Potapov publications, especially those in 2019.

*Reviewer #2 (Recommendations for the authors):*

The abstract section seems to consist mostly of the research background and of the authors' interpretations (L17-26). It could be improved by describing the isotopic results and patterns more clearly.

L80: Please consider explaining what "generic assumption" means here.

L90: Please change "15N concentration" to N isotope ratio.

L129: I think "stability" could not be tested in the present study.

L162: Please provide a list showing what kinds of taxa were used in this study.

L176: I suppose that the up-loaded file Table S1 may differ from what the authors intended. Please check the file.

L186: Were Collembola, Oribatida, and Chilopoda collected at the same sampling occasion with this study?

L310: It is difficult for my eyes to see the size of the points (Figure 1).

L393: I guess that the correlations among environmental factors have been reported previously by the authors' group. If so, please consider omitting them and simplifying this figure.

L445-448: I think this sentence clearly shows the necessity to consider the taxonomic group included in the samples to interpret the isotopic values accurately.

L534: The range of d15N does not necessarily reflect the length of food chain. As the authors explain (L528-529), the primary producers (vascular plants, algae, and lichens) have different 15N values in the study sites, which should have affected the range of 15N of consumers.

---

## [Author Response]

Essential revisions:1) As one of the reviewers noted, the use of stable isotopes on higher taxonomic ranks needs to be fully re-evaluated and much better justified. Since these analyses are critical to supporting the study's claims, this point requires careful revisions.

Indeed, the use of stable isotopes on higher taxonomic ranks is unusual since species within groups may have different stable isotope niches. However, we believe that this approach is justified and best suited for our research questions. We added more detail on the justification of the approach to the text as below:

“For stable isotope analysis we adopted a group level analysis representing the stable isotope niche at the level of taxonomic groups. Although this approach may miss the variability in stable isotope niches of species within high-rank taxonomic groups, it has the advantage that it integrates across species allowing generalizations on the trophic structure and energy flux of whole communities. The approach has been recently advocated for analyzing the channeling of energy through food webs using lipid profiling (Kühn et al., 2018), but has not been adopted yet in stable isotope analysis although it has been shown that at least in soil high-rank animal taxa typically represent the trophic niches of species in most taxa (Potapov et al., 2019). To ensure that our samples reliably represent the trophic niche of the studied taxa we included (if ever possible) 15 individuals per taxon per study site. Doing that we considered the turnover of species among sites and focused on dominant species representing the majority of biomass, which we considered most important for our approach.”

Beyond that, having complete representation of species across groups was hardly feasible – in many groups up to 50% of species are new to science and have not been described. There is always a trade-off between being detailed and having a complete community picture. In our study we intend to explore food web structure and function comprehensively across meso-and macrofauna and across 32 distinct sites to get overview of soil food web changes under tropical land use. We believe that we have a good balance by using representative selection of individuals within high-rank taxa.

2) Other comments and requests of expanding the discussion (likely in an appendix) require attention.

The discussion has been modified as suggested. We address all comments in our point-by-point response below.

Reviewer #1 (Recommendations for the authors):The Introduction is already long, but it may benefit from an expanded discussion of previous literature on metabolic rates and soil communities (beyond setting the fast-bacteria and slow-fungi framework). You get to know that this ms deals with metabolic rates of inverts just in the Objectives, and you get to know that metabolic data come from previous literature in M&M.

We added the discussion on metabolic rates and soil communities in the introduction as below:

“Moreover, abundance and biomass each are biased in reflecting the functional role of consumers covering wide body size ranges. While abundance is biased towards the importance of small organisms, biomass is biased towards that of large ones. Considering these limitations, energetic demands of consumers (i.e., metabolic rates) may be used as less biased metric (Brown et al., 2004). In recent years, the energy flux approach was successfully used to represent functional changes in food webs, and therefore to link multitrophic biodiversity to ecosystem functioning (Barnes et al., 2018, 2014; Jochum et al., 2021). To the best of our knowledge, however, the energy flux approach has never been used in conjunction with stable isotope analysis.”

For me, one major shortcoming this manuscript has is the very reduced Discussion of specific taxa. I agree that expanding a Discussion to 23 invertebrate taxa may be too much. But, the manuscript does little to satisfy specific questions arising on specific taxa and thus its ability to generate testable hypothesis remains low. An expanded Discussion in an Appendix, maybe accompanying the graphs already in there, may be useful for this purpose. Just think about the many *single* taxa taxonomists that will read this ms and wish to read more about their preferred taxa. You can use such an appendix to set ideas to test in the future.

We expanded discussion on specific taxa as below in the main text instead of the appendix to make it more visible. We tried to keep it concise while covering as many interesting patterns as possible.

“Collembola, Symphyla and Isopoda showed a much higher 15N enrichment than e.g., Oribatida, but all three groups occupy similar trophic positions in temperate forests and predominantly function as decomposers (Potapov et al., 2019). This difference may be caused by low litter quality in tropical forests forcing decomposers to switch to more microbial or even animal diet (Illig et al., 2005).”

“Among other groups with a wide food spectrum, Diptera had higher isotopic values (both Δ15N and Δ13C) than Coleoptera, indicating that flies are more closely linked to detrital and microbial food chains in tropical soil food webs than beetles. Isopoda had strikingly high Δ15N values for macrodecomposers, possibly due to intense coprophagy (Potapov et al., 2022). Overall, despite general similarities, we also have found consistent differences between tropical and temperate soil food webs. Further studies comparing differences in soil food-web structure and associated soil functions across temperate and tropical ecosystems are needed to prove the generality of these differences and their consequences for biodiversity – ecosystem functioning relationships.”

Please correct the many ways to refer to Mr Potapov publications, especially those in 2019.

Thank you for locating the inconsistency. We once again checked the publications.

Reviewer #2 (Recommendations for the authors):The abstract section seems to consist mostly of the research background and of the authors' interpretations (L17-26). It could be improved by describing the isotopic results and patterns more clearly.

We have incorporated the isotopic results and patterns into the abstract as below:

“Across the 23 animal groups studied, most of the taxa switched to freshly-fixed plant carbon (low Δ^13^C values) indicating ‘fast’ energy channeling in plantations as opposed to 'slow' energy channeling through the detrital pathway in rainforests (high Δ^13^C values). These shifts led to changes in isotopic divergence, dispersion, evenness and uniqueness. However, earthworms as major detritivores stayed unchanged in their trophic niche and monopolized the detrital pathway in plantations, resulting in similar energetic metrics across land-use systems.”

L80: Please consider explaining what "generic assumption" means here.

We deleted this sentence since the message it conveys duplicated in the sentence that follows.

L90: Please change "15N concentration" to N isotope ratio.

Done.

L129: I think "stability" could not be tested in the present study.

Yes, stability is indeed not easy to be tested in this study, and we now omitted it from the hypothesis.

L162: Please provide a list showing what kinds of taxa were used in this study.

We added the list of taxa after this sentence.

L176: I suppose that the up-loaded file Table S1 may differ from what the authors intended. Please check the file.

There was a wrong reference, which is now deleted, thank you.

L186: Were Collembola, Oribatida, and Chilopoda collected at the same sampling occasion with this study?

Yes. They are all collected at the same sampling in this study. This is now explicitly stated in the manuscript.

L310: It is difficult for my eyes to see the size of the points (Figure 1).

We made the points less transparent to improve visibility.

L393: I guess that the correlations among environmental factors have been reported previously by the authors' group. If so, please consider omitting them and simplifying this figure.

The correlations among environmental factors and microbial groups were reported by our co-author Dr. Krashevska (https://link.springer.com/article/10.1007/s00374-015-1021-4), and we refer to this paper in the Materials and methods. However, the relationships between community and energetic food web metrics, and environmental factors stay new. The correlations among environmental factors are displayed to explain the selection of the environmental factors for the SEM (Figure 6). Therefore, we suggest to keep the figure in the present form, but now we also refer to the study of Krashevska et al., (2015) in the caption of the figure.

L445-448: I think this sentence clearly shows the necessity to consider the taxonomic group included in the samples to interpret the isotopic values accurately.

At our study sites, Japygidae were by far more abundant than other Diplura (e.g., Campodeidae). We realise that merging these two groups is a simplification, but we believe that our analysis is representative to assess core functional groups at our sites. Our overarching response to this point can be found above in ‘Essential Revisions’.

L534: The range of d15N does not necessarily reflect the length of food chain. As the authors explain (L528-529), the primary producers (vascular plants, algae, and lichens) have different 15N values in the study sites, which should have affected the range of 15N of consumers.

We agree that the variation of Δ^15^N values in primary producers among land-use systems may also affect the range of Δ^15^N, but jungle rubber also had the highest non-calibrated maximum Δ^15^N values among the four systems. Therefore, we kept our discussion but now also discuss other factors affecting the range of Δ^15^N in this paragraph as below:

“As a note of caution, the δ^15^N values of primary producers (vascular plants, algae, and lichens) may vary among our study systems, which may have affected the Δ^15^N values of consumers, but unlikely our overall conclusions.”